# SSNIP-seq: A simple and rapid method for isolation of single-sperm nucleic acid for high-throughput sequencing

Stevan Novakovic[1], Vanessa Tsui[1,2], Tim Semple[3], Luciano Martelotto[3], Davis J. McCarthy[4,5], Wayne Crismani[1,2]*

**1** DNA Repair and Recombination Laboratory, St Vincent's Institute of Medical Research, Melbourne, Victoria, Australia, **2** The Faculty of Medicine, Dentistry and Health Science, The University of Melbourne, Melbourne, Victoria, Australia, **3** Single Cell Innovation Laboratory, Centre for Cancer Research, The University of Melbourne, Melbourne, Victoria, Australia, **4** Bioinformatics and Cellular Genomics, St. Vincent's Institute of Medical Research, Melbourne, Victoria, Australia, **5** Melbourne Integrative Genomics, Faculty of Science, The University of Melbourne, Melbourne, Victoria, Australia

* wcrismani@svi.edu.au

**Data Availability Statement:** The protocol is on protocols.io (dx.doi.org/10.17504/protocols.io.6qpvr67jbvmk/v1). https://www.protocols.io/view/ssnip-seq-a-simple-and-rapid-method-for-

## Abstract

We developed a simple and reliable method for the isolation of haploid nuclei from fresh and frozen testes. The described protocol uses readily available reagents in combination with flow cytometry to separate haploid and diploid nuclei. The protocol can be completed within 1 hour and the resulting individual haploid nuclei have intact morphology. The isolated nuclei are suitable for library preparation for high-throughput DNA and RNA sequencing using bulk or single nuclei. The protocol was optimised with mouse testes and we anticipate that it can be applied for the isolation of mature sperm from other mammals including humans.

## Introduction

### Background

Sperm can be used with genomic technologies to study diverse topics such as fertility, genome structure and forensic science [1–5]. A major challenge for the isolation of individual haploid cells from males for sequencing is that they stick to most pipette tips and tubes, and traditional methods for bulk sperm isolation use technically challenging density gradients. Therefore, the motivation for developing this technique is to have a simple and rapid nucleic acid preparation technique that is compatible with high-throughput sequencing technologies, including for bulk and single cell ATAC-seq, RNA-seq and similar approaches. Several studies have been published that focused on the sequencing of individual mouse or human sperm [6–11] highlighting the interest in the biology and the utility of a simple method that uses standard laboratory materials and equipment.

### Development and application of the method

We have developed a simple and reliable protocol for the isolation of over 100,000 haploid nuclei derived from mouse testis, which we refer to as SSNIP-seq (single-sperm nuclei

isolation-6qpvr67jbvmk/v1 All relevant data are within the paper and its Supporting information files.

**Funding:** WC and DJM received funding related to this work from the Australian National Health and Medical Research Council (GNT1129757, GNT1185387). WC is a fellow of the Victorian Cancer Agency (MCRF21006).

**Competing interests:** The authors have declared that no competing interests exist.

isolation protocol). The protocol is adapted from the 'Frankenstein' protocol for nuclei isolation [12] and makes modifications that have allowed us to reproducibly isolate over 100,000 haploid nuclei from a single mouse testis or epididymis in approximately 30 minutes (Fig 1, S1 Table). We use 100,000 as a minimum target number in this protocol as droplet-based single-nucleus library preparation methods can require a high concentration of nuclei in a small volume. This suspension is then used as the starting material for the protocol. It is important to include a diploid control sample. This control is used to define what is a diploid cell at G1 or G2 during cell sorting, and in turn what cells are haploid. This method is developed to isolate nuclei from both spermatids and spermatozoa. We use spleen as high numbers of cells can be put into suspension with very simple mechanical disruption. The resulting nuclei from all samples are stained with DAPI and sorted using flow cytometry to isolate the haploid nuclei from the testis sample, which can then be used as material for a variety of high-throughput sequencing techniques (Fig 1). We have successfully used these nuclei in bulk ATAC-seq, scATAC-seq and single nuclei CNV experiments for mouse haploid nuclei [13]. We propose that the protocol can be easily adapted for mature human sperm.

## Materials and methods

### Equipment and software

- Cell sorter (BD400 Class II FACSAria Fusion Cell Sorter with FACS Diva software, Becton Dickinson)

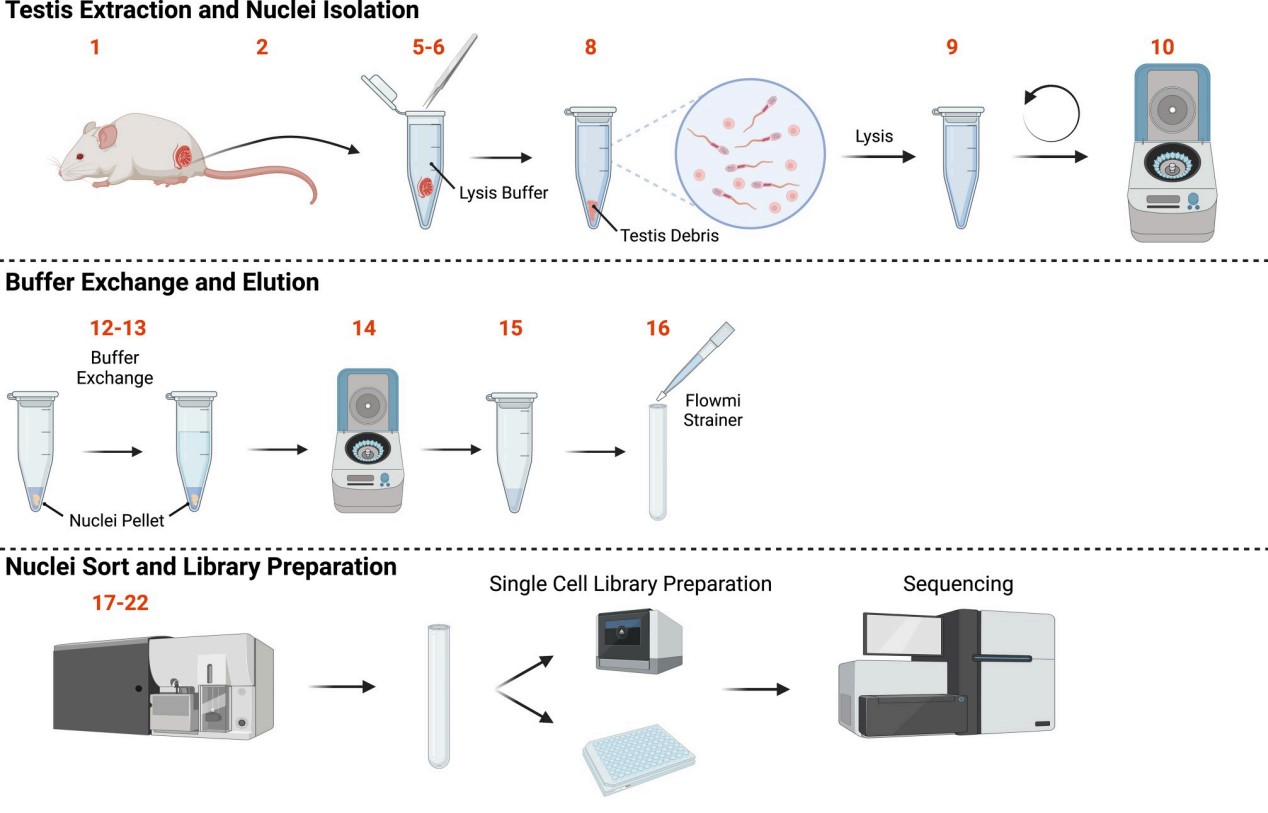

**Fig 1. Schematic summary of the protocol.** The protocol workflow consists of three key parts: 1) tissue collection, 2) nuclei isolation and buffer exchange, and 3) cell sorting. This protocol results in the isolation of haploid sperm-derived nuclei which may then be utilised in a variety of downstream applications, including both plate- and droplet-based approaches.

- Immunofluorescence microscope that can visualise DAPI

- Benchtop microcentrifuge with temperature control or located in a cold room. We use an Eppendorf Centrifuge 5418 in a 4 °C cold room.

- PCR tube adapters for the microcentrifuge

- Timer

## Laboratory materials

- Flowmi cell strainers, 40 μm (Bel-Art, cat no. H13680-0040)

- DNA LoBind microcentrifuge tubes, 1.5 mL (Eppendorf, cat no. 0030 108 418)

- Round-bottom polypropylene tubes with cap, 5 mL (Falcon, cat no. 352063)

- Wide bore pipette tips, 1000 μL (Axygen, cat no. T-1005-WB-C) (optional)

- Micropipettes DV1000, DV200, DV20, D10 (THL, cat no. PZ-7901)

- Cell strainers, 70 μm (Corning, cat no. 431751)

- Tweezers with pointed tips, 115mm (Weller Erem, cat no. 5SA)

- Tissue-culture treated culture dishes, 35 x 10 mm (Corning, cat no. 430165)

- Thin wall PCR tubes with flat cap, 200 μL (Axygen, cat no. PCR-02-L-C)

- 96-well plate with round bottom (Greiner, cat no. 650101)

- Sterile syringe 0.2 μm filters, acrodisc (Pall, cat no. 2415600)

- 50 mL syringe (BD, cat no. 300144)

- 50 mL centrifuge tube (Sarstedt, cat no. 92.547.254)

- 2 mL plastic syringe (BD, cat. No. 302204)

## Biological material

One wild-type male mouse six weeks or older. This protocol was developed using C57BL/6J, FVB/N, and F1(C57BL/6J x FVB/N) mice.

## Solutions

- 10x Dulbecco's Phosphate Buffered Saline (DPBS), without calcium chloride and magnesium chloride (Sigma Aldrich, cat no. D1408-500ML)

- 1x DPBS, without calcium chloride and magnesium chloride (Sigma Aldrich, cat no. D8537-500ML)

- 4, 6-Diamidino-2-phenylindole dihydrochloride (DAPI; Sigma Aldrich, cat no. D8417-1MG)

- 'Bovostar' Bovine Serum Albumin (BSA; Bovogen Biologicals, cat no. BSASAU-0.1)

- Nuclei EZ Lysis Buffer from Nuclei EZ Prep Kit (Sigma Aldrich, cat no. NUC101-1KT)

## Buffers and reagents

- **DAPI stock**–Prepare 1 mg / mL of DAPI in sterile water. Filtered using a 0.22 μm filter and stored at -20˚C.

- **10% BSA**–Prepare a 10% BSA (w/v) stock solution with sterile water. Filter using a 0.22 μm filter and store at 4˚C for up to a month.

- **Nuclei Wash Resuspension Buffer (NWRB) with 1% or 0.1% BSA (w/v)**–Prepare solution using 10x DBPS, 10% BSA stock and sterile water. Make fresh prior to use and keep on ice.

- **NWRB with 1% BSA and DAPI (10 μg / mL)**–Prepare using 10x DBPS, 10% BSA stock, DAPI stock and sterile water. Make fresh prior to use, keep on ice and protect from light.

## Procedure

All steps must be completed quickly, and samples kept at 4˚C at all times. The use of Lo-Bind microcentrifuge tubes is critical to avoid excessive loss of sperm cells or nuclei.

**Tissue isolation.**

1| Sacrifice mice using $CO_2$ or cervical dislocation. Following local SOPs that have been approved by an animal ethics committee. All experimental procedures were approved in writing by the St. Vincent's Hospital Melbourne Animal Ethics Committee.

2| Dissect testis and spleens from each mouse, removing as much fat and unwanted connective tissue as possible, and place each organ individually in a 1.5 mL microcentrifuge tube containing cold DPBS (Fig 1).

- If nuclei isolation is performed on the day, place the testis on ice and begin the protocol within 30 minutes of dissection. Otherwise, the testes can be individually placed in an empty 1.5 mL microcentrifuge tube and snap frozen in liquid nitrogen then stored at -80˚C. We have obtained >100,000 nuclei from frozen testis samples up to 1 month after freezing.

**Spleen cell suspension.**   A spleen cell suspension is formed which will serve as a diploid control.

3| Place a 70 μm cell strainer into a 50 mL centrifuge tube and wet the strainer by pipetting 1 mL cold DPBS onto the strainer mesh.

4| Place the whole spleen into the strainer and homogenise it using the plunger seal of a 2 mL plastic syringe (S1 Fig). Rinse the strainer with an additional 1 mL of cold DPBS. Place the sample on ice until step 7.

- The solution should look homogenous. If the suspension contains clumps of material, it can be filtered into a second 50 mL centrifuge tube using a clean 70 μm cell strainer.

**Nuclei isolation.**   At this stage it is anticipated that you will have:

- Either a fresh or frozen testis;

- If using a frozen testis allow it to thaw on ice for 10 minutes; and

- A spleen to serve as a diploid control

   All steps are performed on ice

5| Place two 1.5 mL microcentrifuge tubes on ice and add 1 mL of chilled Nuclei EZ Lysis Buffer to each.

6| Transfer the fresh or thawed testis to one of the microcentrifuge tubes.

- Release the seminiferous tubules from the testis by gently squeezing the testis with pointed tweezers until the testis burst. Gently tear and homogenise the testis and in turn the seminiferous tubules several times with the tweezers to further break them apart and allow the release of more cells into solution. The solution should be cloudy due to the high cell number.

7| Add 300 μL of the spleen cell suspension (from step 4) to the second microcentrifuge tube (Fig 1).

8| Incubate both the spleen and the testis samples on ice for 5 minutes to allow cell lysis and the release of nuclei. From here until step 18 (Cell sorting) both the testis and spleen sample are treated the same.

- After 3 minutes of the 5-minute incubation gently invert the samples two or three times to allow better mixing of the cells with the Nuclei EZ Lysis Buffer.

- For mutants with smaller testis or low sperm production it may be necessary to break the testis apart more extensively or break apart two testes in the same tube. To overcome volume constraints this may need to be performed in a 2 mL Lo-Bind microcentrifuge.

9| Just prior to the end of the 5-minute incubation period, remove any large pieces of non-degraded testis debris.

- Failure to remove debris will result in unwanted clumping of debris and the target nuclei onto the bottom of the tube.

10| Centrifuge the samples at 500 x g for 5 minutes at 4˚C. Remove the supernatant, leaving behind just enough to cover the pellet. Do not allow the pellet to dry.

11| Add 1 mL of Nuclei EZ Lysis Buffer. Repeat steps 8–10.

12| Very slowly add 1 mL of NWRB with 1% BSA and incubate the sample on ice for 5 minutes to allow buffer exchange. Take care to avoid resuspending the pellet.

13| After incubation gently resuspend the pellet by either inverting 5 times or pipetting up and down 5 times with a wide-bore tip.

14| Centrifuge at 500 x g for 5 minutes at 4˚C. Remove the supernatant, leaving behind just enough to cover the pellet. Do not dry pellet.

15| Resuspend the pellet in 300 μL NWRB with 1% BSA and 3 μL of the DAPI stock (final concentration 10 μg / μL) by either inverting 5 times or pipetting up and down 5 times with a wide-bore tip.

- To resuspend the pellet gently invert the sample several times. If it fails to resuspend a 1 mL wide-bore pipette tip can be used to gently pipette mix the sample up and down up to 10 times

- Full resuspension of the pellet is not essential and, in our hands, still yields excess of 100,000 haploid nuclei. Excessive inverting or mixing may damage the integrity of the nuclei.

16| Filter the sample using a 40 µm Flowmi cell strainer into a 5 mL polypropylene tube which is kept on ice and in the dark.

**Cell sorting and cytometer settings.** Cytometric analysis should be performed as soon as possible; ideally within 10–30 minutes post-isolation. It is recommended to work with an experienced cytometric operator to optimise the sorting procedure under your laboratory's conditions. We recommend using a 75 µm nozzle, and a flow rate of fewer than 10,000 events per second. It is critical to optimise your sort for purity rather than yield. Refer to your instrument manufacturers setup and operation guide.

Prior to nuclei sorting turn on the cytometer as per the manufacturer's instructions. Use a collection device capable of holding 5 mL polypropylene tubes and ideally maintaining 4˚C.

Diploid control material–here, the single splenic nuclei suspension–is used to identify the G1 diploid peak based on DAPI intensity, which in turn is used to identify haploid cells from reproductive tissue (Fig 2). This G1 peak–of the diploid sample–is placed in-range, visually, on the x-axis by adjusting the voltage. Haploid sperm nuclei can then be distinguished from diploid nuclei using the same gating parameter but with a final DAPI intensity approximately half that of the diploid control.

17| Preparation of FACS collection tubes and plates–For each sample add 200 µL of NWRB with 0.1% BSA in a 5 mL polypropylene tube. For collections in 96-well plates add 50 µL of NWRB with 0.1% BSA.

18| Critical step–Set the sorter precision to '4-Way Purity', this will ensure that the final volume of the sorted nuclei is kept to a minimum (~1 nucleus sorted per nL), preventing dilution of the sample with FACS sheath solution.

- On some instruments this setting may be called 'Single Cell' precision when sorting in 96-well plates.

- It is preferable to use slow flow rates, but a medium flow rate can be used if necessary.

19| Using the standard workflows of your FACS facility, gate for individual nuclei (Fig 2A–2E). Gate the main population of nuclei using linear forward scatter area (FSC-A-lin) and linear side scatter area (SSC-A-lin) by region gate 1 (R1) (Fig 2A). Then gate single cells R2, R3, R4, R5 (Fig 2B–2E).

20| To discriminate haploid from diploid nuclei (R6), use the DAPI stain to quantify the DNA content of each nucleus (Fig 2F). Splenic nuclei serve as diploid controls here, which are essential to confidently gate around the peak that corresponds to the DAPI-stained haploid nuclei (Fig 2F). It is therefore important to set the voltage of the laser that excites DAPI allow visualisation of the G1 peak of the diploid sample, as well as a haploid peak, in the same range on the x-axis (Fig 2F, right). The haploid peak–on the x-axis of a histogram–will be half that of the diploid G1 peak.

21| Diploid nuclei control sorting–Sort the target number of diploid nuclei–typically 100,000 or greater–into the previously prepared 5 mL polypropylene tube or 96 well plate.

- While this diploid control sample is generally not used for library preparation and sequencing, it is helpful to perform sorting of individual nuclei from this sample as it is typically easier to isolate 100,000 splenic nuclei than 100,000 haploid nuclei from testis while establishing the protocol.

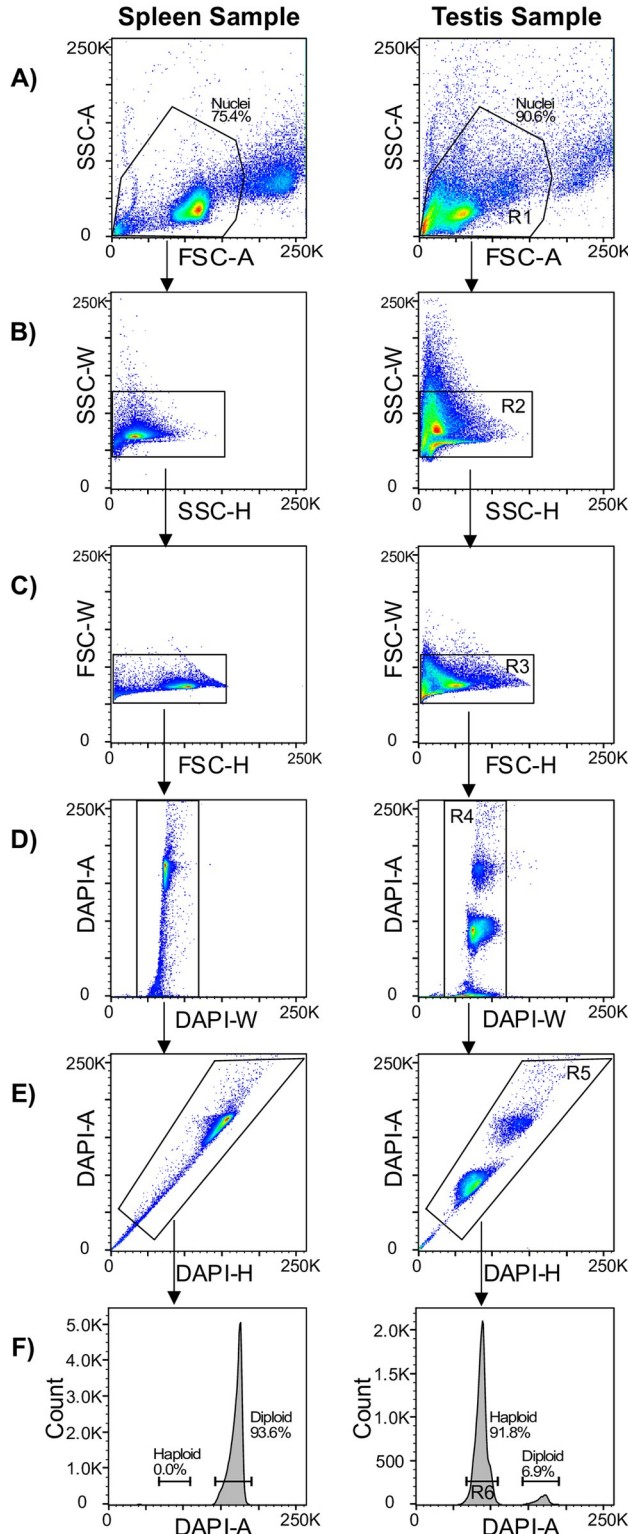

**Fig 2. Flow cytometric gating strategy for the isolation of haploid cells.** The following gating steps are applied sequentially, as indicated by the arrows i.e. all events in 'B' represent only the gated events from 'A' and so on to 'F'. **(A)** Nuclei are isolated from the rest of the population initially using linear forward scatter area (FSC-A-lin) and linear side scatter area (SSC-A-lin) by region gate 1 (R1). **(B)** Single nuclei are isolated using: region gate 2 (R2) on linear side scatter width (SSC-W-lin) and linear side scatter height (SSC-H-lin); **(C)** region gate 3 (R3) on linear forward scatter

width (FSC-W-lin) and linear forward scatter height (FSC-H-lin); **(D)** region gate 4 (R4) on linear DAPI area (DAPI-A-lin) and linear DAPI width (DAPI-W-lin); and **(E)** region gate 5 (R5) on linear DAPI area (DAPI-A-lin) and linear DAPI height (DAPI-H-lin) **(F)** The use of a diploid control allows flow cytometric gating of haploid sperm. Diploid samples such as splenic cells used here, will have a G1 (2c DNA content) and a G2 peak (4c DNA content, out of range in this figure) peak respectively. Acquire events from region 6 (R6), the haploid nuclei which have a 1c DNA content (R6) which are not present in the diploid control sample. The values on all both axes of panels A-E and the x-axis of panel F are arbitrary values that represent increasing intensity of signal from 0, the lowest, to 250K the greatest. Major and minor ticks represent 50K and 10K increments respectively.

22| Haploid sample sorting–Sort the target number of haploid nuclei–typically 100,000 in our experiments–into the previously prepared 5 mL polypropylene tube or 96 well plate.

- The time required to isolate 100,000 cells is dependent on the concentration of nuclei in solution, but typically splenic and testicular samples require 2 minutes to reach this target.

- If material is very limiting, it might be helpful to consider a plate-based method to maximise yield and therefore to sort one nucleus per well of an appropriate plate at this step.

**Post-FACS concentration of nuclei (for droplet-based methods).**

23| Centrifuge the samples at 500 x g for 5 min at 4˚C. Remove the supernatant, leaving approximately 50 μL. Do not allow the pellet to dry.

At this point the nuclei can be used for a variety of applications, including single- and bulk-nuclei sequencing.

The specific application will determine what buffer the nuclei should be exchanged into. Below we provide a case study where we isolated haploid sperm nuclei for library preparation with the droplet-based 10x Genomics single-nucleus ATAC-seq kit.

**Case study: 10x single-nucleus ATAC-seq.**

24| Add 300 μL NWRB with 0.1% BSA and 0.01% Digitonin and incubate on ice for 5 minutes.

25| Centrifuge at 500 x g for 5 min at 4˚C. Remove the supernatant, leaving behind ~50 μL. Do not dry the pellet.

26| Add 300 μL Dilute Nuclei Buffer (10x Genomics) and gently resuspend the nuclei by pipetting up and down 3 times using a wide-bore pipette tip.

27| Centrifuge at 500 x g for 5 min at 4˚C. Remove the supernatant, leaving behind ~50 μL. Do not allow the pellet to dry.

28| Gently resuspend the nuclei by pipetting up and down 3 times using a wide-bore pipette tip.

29| Quantify the concentration of the nuclei suspension using a hemocytometer. We use a Leica Thunder under 40x magnification, with brightfield to locate the hemocytometer grid and fluorescence to identify the DAPI-stained nuclei. This also allows use to check the integrity of the nuclei (Fig 3).

- For droplet-based methods of partitioning single nuclei, such as the 10x single-nucleus kits, all nuclei must be intact.

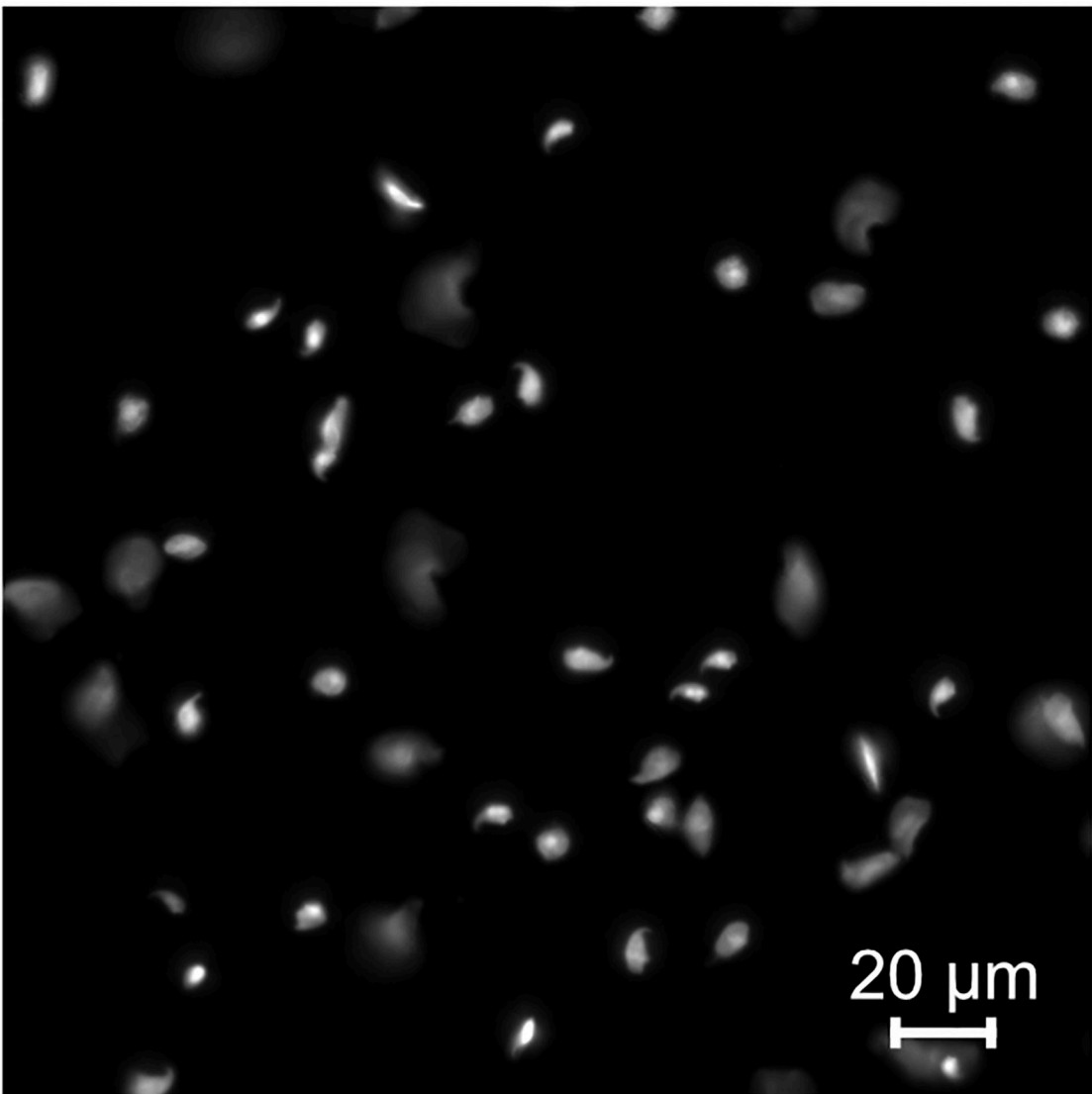

**Fig 3. Intact haploid sperm and spermatids are used as input for droplet-based or plate-based high-throughput sequencing library construction.** Representative image of high-quality intact haploid cells isolated using the protocol.

- We have used this method and reliably achieved a final concentration of 400 to 450 nuclei per µL, with the aim of loading 1500 nuclei into a 10x Chromium.

**Next steps for library preparation.**

30| Follow the manufacturer's protocol for library preparation. From here we have made high-quality "single-nucleus" and "bulk" libraries with the haploid nuclei with the Next GEM Single Cell ATAC Library Kit v1.1 (10x Genomics) and with Nextera (Illumina) DNA Library Prep Kit respectively. For bulk sequencing we aim to sequence 50,000 nuclei.

**Sequencing and bioinformatics considerations.**

31| The sequencing depth required per cell will vary according to experimental aims. We have found in the early stages of protocol development that a useful guide can be to sequence a

given sample in two steps. First, request your sequencing platform obtain approximately 200 million paired-end reads per one thousand expected cells. Once this data has been analysed, and cell number and coverage is approximated, a second round of sequencing is performed to obtain the required depth per cell.

- If the recovery of total number of nuclei recovered is lower than expected, in the first instance it is important to independently assess how many cells are present by assessing the number of barcodes with > 800,000 high-quality paired-end reads that map to the target genome. From here the threshold can be adjusted based on the researchers' decisions.

- Next time the entire experiment is performed, the number of nuclei used for library preparation can be doubled as nuclei concentration should not be limiting.

- When using bulk- or single-nucleus ATAC-seq, the nuclei permeabilization step with digitonin, or similar detergents, and the associated buffers can be re-optimised.

The protocol described in this peer-reviewed article is published on protocols.io (dx.doi.org/10.17504/protocols.io.6qpvr67jbvmk/v1) and is included for printing purposes as S1 File.

**Variations of the protocol for other applications.** The protocol can be adapted to work with plate-based methods. Steps 1–21 should provide a sensible starting point to isolate high-quality haploid nuclei. Depending on the desired library preparation method (e.g. ATAC or whole-genome amplification) user-specific modifications can be made at the appropriate steps.

**Troubleshooting.** If poor yields are obtained or nuclei do not appear intact, we suggest optimising the salt concentrations of the PBS. The tonicity that is ideal for mouse and human cells can differ and may be the cause of cases of cytolysis.

Further, the visualisation of cells and nuclei at every stage is possible with bright field microscopy or immunofluorescence with DAPI. Visualisation of sample integrity and concentration should be used routinely to establish the protocol, and when problems occur, to identify at which stage cells or nuclei are lost or perturbed.

## Anticipated results

The anticipated results with the haploid sperm nuclei isolation protocol depend on the mouse genetic background and age. The protocol was developed with wild-type mice and mice with mutations in genes required for spermatogenesis may affect the yield of this protocol. The successful procedure is dependent on working proficiently with the mice and their biological material, the composition of reagents, the materials and proficient use of the cell sorter. Upon visualisation of DAPI-stained haploid nuclei a distinct sperm head shape should be seen. No extruded chromosomes should be visible as this will result in cross contamination between cells when analysing sequencing results.

## Supporting information

**S1 Fig. Homogenisation of splenic cells as diploid controls.** Cartoon representation of how splenic cells are obtained via homogenisation and filtration.
(TIFF)

**S1 Table. Yields of intact single haploid nuclei with the protocol.** Yield data from our laboratory using the protocol since optimisation. The data is from fresh and frozen testes, fresh epididymides, and cover wild-type and hypospermatogenic (*Fancm*-deficient) mice. "gated haploid nuclei" represents the total number of haploid nuclei that were gated using the strategy shown in Fig 2A–2F and isolated at gate "R6" (Fig 2F). "haploid percent of population (events

pre-gating)" represents the number of haploid nuclei were gated (R6, Fig 2F) for a given experiment divided by the total number of events detected pre-gating. i.e. (R6 / events pre-gating) X 100.
(XLSX)

**S1 File. Step-by-step protocol, also available on protocols.io.**
(PDF)

## Acknowledgments

We wish to thank Anthony Di Carluccio for assistance with flow cytometry and cell sorting.

## Author Contributions

**Conceptualization:** Stevan Novakovic, Vanessa Tsui, Tim Semple, Luciano Martelotto, Davis J. McCarthy, Wayne Crismani.

**Formal analysis:** Stevan Novakovic, Davis J. McCarthy.

**Funding acquisition:** Wayne Crismani.

**Methodology:** Stevan Novakovic, Vanessa Tsui, Tim Semple, Luciano Martelotto.

**Supervision:** Wayne Crismani.

**Writing – original draft:** Stevan Novakovic, Wayne Crismani.

**Writing – review & editing:** Stevan Novakovic, Vanessa Tsui, Tim Semple, Luciano Martelotto, Davis J. McCarthy, Wayne Crismani.

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
