## [Decision Letter · Decision Letter 0]

12 Jul 2022

PONE-D-22-14844A simple and rapid method for isolation of single-sperm nucleic acid for high-throughput sequencingPLOS ONE

Dear Dr. Wayne Crismani,

Thank you for submitting your manuscript to PLOS ONE. After careful consideration, we feel that it has merit but does not fully meet PLOS ONE’s publication criteria as it currently stands. Therefore, we invite you to submit a revised version of the manuscript that addresses the points raised during the review process.

We look forward to receiving your revised manuscript.

Kind regards,

Joël R Drevet, Ph.D.

Academic Editor

PLOS ONE

Journal Requirements:

2. We note you have not yet provided a protocols.io PDF version of your protocol and a protocols.io DOI. When you submit your revision, please provide a PDF version of your protocol as generated by protocols.io (the file will have the protocols.io logo in the upper right corner of the first page) as a Supporting Information file. The filename should be S1_file.pdf, and you should enter “S1 File” into the Description field. Any additional protocols should be numbered S2, S3, and so on. Please also follow the instructions for Supporting Information captions [https://journals.plos.org/plosone/s/supporting-information#loc-captions]. The title in the caption should read: “Step-by-step protocol, also available on protocols.io.”

Please assign your protocol a protocols.io DOI, if you have not already done so, and include the following line in the Materials and Methods section of your manuscript: “The protocol described in this peer-reviewed article is published on protocols.io (https://dx.doi.org/10.17504/protocols.io.[...]) and is included for printing purposes as S1 File.” You should also supply the DOI in the Protocols.io DOI field of the submission form when you submit your revision.

If you have not yet uploaded your protocol to protocols.io, you are invited to use the platform’s protocol entry service [https://www.protocols.io/we-enter-protocols] for doing so, at no charge. Through this service, the team at protocols.io will enter your protocol for you and format it in a way that takes advantage of the platform’s features. When submitting your protocol to the protocol entry service please include the customer code PLOS2022 in the Note field and indicate that your protocol is associated with a PLOS ONE Lab Protocol Submission. You should also include the title and manuscript number of your PLOS ONE submission.

WC and DJM received funding related to this work from the Australian National Health and Medical Research Council (GNT1129757, GNT1185387). 

 WC and DJM received funding related to this work from the Australian National Health and Medical Research Council (GNT1129757, GNT1185387).

6. Please ensure that you refer to Figure 2 in your text as, if accepted, production will need this reference to link the reader to the figure.

Additional Editor Comments:

Both expert reviewers in the field found the protocol interesting and that it could be a good addition to the panel of available techniques. However, they both ask for several clarifications that need to be considered. Please revise accordingly.

Reviewers' comments:

Reviewer's Responses to Questions

**Comments to the Author**

1. Does the manuscript report a protocol which is of utility to the research community and adds value to the published literature?

Reviewer #1: Yes

Reviewer #2: Yes

2. Has the protocol been described in sufficient detail?

Descriptions of methods and reagents contained in the step-by-step protocol should be reported in sufficient detail for another researcher to reproduce all experiments and analyses. The protocol should describe the appropriate controls, sample sizes and replication needed to ensure that the data are robust and reproducible.

Reviewer #1: No

Reviewer #2: Yes

3. Does the protocol describe a validated method?

Reviewer #1: Yes

Reviewer #2: No

4. If the manuscript contains new data, have the authors made this data fully available?

Reviewer #1: N/A

Reviewer #2: No

**5. Is the article presented in an intelligible fashion and written in standard English?**

Reviewer #1: Yes

Reviewer #2: **No: **It is a clear and concise description, I noticed only some minor typos:

line 127 – placed

lines 151/152 – squeezing the testis with pointed tweezers until the testis bursts

line 156 – microcentrifuge tube

line 192 – Not a functional sentence

6. Review Comments to the Author

Reviewer #1: The paper by Novakovic et al present a method for the isolation of haploid nuclei from fresh and frozen mice testis. The methods can be applied for several purposes as indicated by the Authors, who also anticipate that the method could be used also for human specimens.

The method is of interest, however there are some points needing detailing and clarification.

Specific points:

1. The Authors do not report any data on the yield either of the global method or of the cell sorting step, whereas it appears an important detail. It appears that a huge number of haploid nuclei is lost as the method uses one testis and recoveries 100,000 (or 300,000?, line 185). Thus this method could be used for only certain applications. Have the Authors any explanation for such high cell lost? This point is important because if the method used human specimens, it is expected that the available material would be much smaller and the question is whether the starting material was sufficient with this very low yield.

2. Cell sorting. Details on strategy to sort haploid nuclei is completely lacking. Which gate in FSC/SSC dot plot? Did you exclude doublets? Which were the technical settings of the instrumentation?

3. Line 213. The haploid peak is represented in a fluorescence distribution histogram. What do you mean with density plot? In addition, use “gate around peak” not “gate with”.

4. In the abstract, the Author claim that the method is reproducible, but non data on this point are provided

Minor:

Figure 1 is confused with Figure 2

Referring to real Figure 1 is lacking in the text

Line 195, G1 peak does not seem centred on x-axis.

Reviewer #2: The protocol provides a valuable addition to the field of single cell assays of testis tissue, as especially the nucleus isolation has proven tricky in the past. The description of working steps is concise, and I would feel confident following this protocol.

However, I do have some concerns with the manuscript:

1) The authors refer to the method as “single-sperm nucleus isolation”. I find this misleading. Mouse testis contains haploid spermatids at different stages in development and transcription profiles differ distinguishably from round spermatids to later spermatid stages. The nucleus shown in figure 2C corresponds to a later spermatid stage, however FACS sorting of the haploid population would in my opinion also include more immature round spermatids. If this population is excluded in the protocol, it is unclear how. This should be discussed and Figure 2 should be updated to depict not only a single nucleus, but a representative sample of the obtained haploid nuclei.

2) In line with my first concern, the authors imply that this protocol can be adapted to isolate mature human sperm. Maturation of sperm, including development of fertilisation capability and motility, is completed not in the testis, but in the epididymides. In my experience, motile sperm is “more sticky” than spermatids and it would be interesting to see how the protocol performs with this population. At least, the distinction between spermatids, spermatozoa and mature sperm should be made clear in the manuscript and discussed in more detail.

3) I was unable to assess the data from the case study, as it was not made available to reviewers prior to publication. Thus, I feel a bit uncomfortable to classify this as a validated method. At least the quality control data for the case study, for example an image of isolated nuclei in the hemocytometer to assess nuclear integrity, should be included.

In light of these points, I recommend the manuscript be revised.

7. PLOS authors have the option to publish the peer review history of their article (what does this mean?). If published, this will include your full peer review and any attached files.

Reviewer #1: No

Reviewer #2: No

---

## [Author Response · Author response to Decision Letter 0]

29 Jul 2022

Our responses are pasted below. However, we have attached this response to reviewers as a word document, which contains formatting which we believe will improves the readability of the responses.

Dear Dr Drevet

We thank you and the reviewers for your thoughtful reading of our manuscript and your helpful comments. The reviewers are complimentary on the quality and value of the protocol e.g. both reviewers confirmed that the protocol “is of utility to the research community and adds value to the published literature”, and “The protocol provides a valuable addition to the field of single cell assays of testis tissue, as especially the nucleus isolation has proven tricky in the past. The description of working steps is concise, and I would feel confident following this protocol.” We address their questions, and all suggestions, below. 

*Please note that references to line numbers in the manuscript are for the clean (without tracked changes) version of the revised manuscript.

Academic Editor’s Report and Journal Requirements:

2. We note you have not yet provided a protocols.io PDF version of your protocol and a protocols.io DOI. When you submit your revision, please provide a PDF version of your protocol as generated by protocols.io (the file will have the protocols.io logo in the upper right corner of the first page) as a Supporting Information file. The filename should be S1_file.pdf, and you should enter “S1 File” into the Description field. Any additional protocols should be numbered S2, S3, and so on. Please also follow the instructions for Supporting Information captions [https://journals.plos.org/plosone/s/supporting-information#loc-captions]. The title in the caption should read: “Step-by-step protocol, also available on protocols.io.”

Please assign your protocol a protocols.io DOI, if you have not already done so, and include the following line in the Materials and Methods section of your manuscript: “The protocol described in this peer-reviewed article is published on protocols.io (https://dx.doi.org/10.17504/protocols.io.[...]) and is included for printing purposes as S1 File.” You should also supply the DOI in the Protocols.io DOI field of the submission form when you submit your revision.

If you have not yet uploaded your protocol to protocols.io, you are invited to use the platform’s protocol entry service [https://www.protocols.io/we-enter-protocols] for doing so, at no charge. Through this service, the team at protocols.io will enter your protocol for you and format it in a way that takes advantage of the platform’s features. When submitting your protocol to the protocol entry service please include the customer code PLOS2022 in the Note field and indicate that your protocol is associated with a PLOS ONE Lab Protocol Submission. You should also include the title and manuscript number of your PLOS ONE submission.

WC and DJM received funding related to this work from the Australian National Health and Medical Research Council (GNT1129757, GNT1185387). 

 WC and DJM received funding related to this work from the Australian National Health and Medical Research Council (GNT1129757, GNT1185387).

This statement has been removed from the Acknowledgements. 

The full ethics statement is now included on lines 140-141 and reads:

“All experimental procedures were approved in writing by the St. Vincent’s Hospital Melbourne Animal Ethics Committee.”

6. Please ensure that you refer to Figure 2 in your text as, if accepted, production will need this reference to link the reader to the figure.

Figure 2 is now referenced appropriately. 

Reviewers' comments:

Reviewer #1: 

The paper by Novakovic et al present a method for the isolation of haploid nuclei from fresh and frozen mice testis. The methods can be applied for several purposes as indicated by the Authors, who also anticipate that the method could be used also for human specimens.

The method is of interest, however there are some points needing detailing and clarification.

Specific points:

1. The Authors do not report any data on the yield either of the global method or of the cell sorting step, whereas it appears an important detail. It appears that a huge number of haploid nuclei is lost as the method uses one testis and recoveries 100,000 (or 300,000?, line 185). Thus this method could be used for only certain applications. Have the Authors any explanation for such high cell lost? This point is important because if the method used human specimens, it is expected that the available material would be much smaller and the question is whether the starting material was sufficient with this very low yield.

We appreciate that this warrants clarification, both in the manuscript and here in this response. We have included a new Supplementary Table 1 which has data for all experiments that used this technique since it was optimised. We have also referenced a new manuscript which is on Biorxiv and under review at PNAS, which has used this technique to identify reproducible phenotypic differences between genotypes (Tsui et al, Figure 1b, https://doi.org/10.1101/2022.06.16.496499).

From a practical perspective, the protocol has been optimised for efficiency – less than an hour – and stringent quality of intact nuclei as opposed to yield. We typically set a target number of haploid cells and stop sorting once that has been reached. E.g. in a very simple example if a user wanted to use a plate-based library preparation approach, and the researcher were using a 384-well plate, they would only require 384 nuclei; one in each well. With respect to numbers such as “100,000” frequently mentioned in the original submission, this stems from aiming to obtain sequencing data for 1,000 nuclei from a droplet-based method (e.g. with 10x Genomics library preparation kits). And to prepare the sequencing libraries with an appropriate concentration of nuclei, based on constraints set by these kits and “droplet encapsulation rates”, we aim to sort 100,000 haploid cells. We have aimed to not place excessive focus on 10x kits as the intended downstream application of this method, but we appreciate that further clarification of where the target number of nuclei originates is required. 

We clarify on lines 61-63; 

- “We use 100,000 as a minimum target number in this protocol as droplet-based single-nucleus library preparation methods can require a high concentration of nuclei in a small volume.”

and lines 307-310.

- “For droplet-based methods of partitioning single nuclei, such as the 10x single-nucleus kits, all nuclei must be intact. We have used this method and reliably achieved a final concentration of 400 to 450 nuclei per μL, with the aim of loading 1500 nuclei into a 10x Chromium.”

From a technical perspective, using this protocol, we can routinely sort more than 300,000 haploid nuclei from a single mouse testis. However, due to our strict selection criteria for single cells (on FACS), approximately 50% of the total sorted nuclei are gated out. As our intention is to sequence individual nuclei, often with droplet-based library methods which have steps where all nuclei are in solution adjacent to one another from protocol steps 23-30 inclusive, it is essential that all nuclei included are intact. If one nucleus included is not intact and extrudes its contents into solution, it can contaminate the data of an unknown quantity of other nuclei when they are encapsulated with the “unwanted” free-floating nucleic acid. This need for the integrity of all nuclei is referred to on lines 307-308 and reads: 

“For droplet-based methods of partitioning single nuclei, such as the 10x single-nucleus kits, all nuclei must be intact.” 

If starting material is very low, a user can consult the newly added Supplementary Table 1 to consider this protocol could be suitable for their needs. Also, sorting individual nuclei into a plate is probably more appropriate in that situation. This comment is added to step 22 and now reads:

“If material is very limiting, it might be helpful to consider a plate-based method to maximise yield and therefore to sort one nucleus per well of an appropriate plate at this step.” 

2. Cell sorting. Details on strategy to sort haploid nuclei is completely lacking. Which gate in FSC/SSC dot plot? Did you exclude doublets? Which were the technical settings of the instrumentation?

We have included a new Figure 2, which illustrates the gating strategy. The gating strategy and technical settings of the instrumentation have been expanded on in in the section “Cell sorting and cytometer settings”, from lines 212-271 and in particular in steps 19-20. This now reads:

Step 19. “Using the standard workflows of your FACS facility, gate for individual nuclei (Figure 2a-e). Gate the main population of nuclei using linear forward scatter area (FSC-A-lin) and linear side scatter area (SSC-A-lin) by region gate 1 (R1) (Figure 2a). Then gate single cells R2, R3, R4, R5 (Figure 2b-e).”

Step 20. “To discriminate haploid from diploid nuclei (R6), use the DAPI stain to quantify the DNA content of each nucleus (Figure 2f). Splenic nuclei serve as diploid controls here, which are essential to confidently gate around the peak that corresponds to the DAPI-stained haploid nuclei (Figure 2f). It is therefore important to set the voltage of the laser that excites DAPI allow visualisation of the G1 peak of the diploid sample, as well as a haploid peak, in the same range on the x-axis (Figure 2f, right). The haploid peak – on the x-axis of a histogram – will be half that of the diploid G1 peak.”

3. Line 213. The haploid peak is represented in a fluorescence distribution histogram. What do you mean with density plot? In addition, use “gate around peak” not “gate with”.

We incorrectly used the term “density plot” where we should have used “histogram”. This has been corrected, and now reads:

“The haploid peak – on the x-axis of a histogram – will be half that of the diploid G1 peak.” [lines 254-255]

We clarified the wording about gates as appropriate too. This now reads:

“Splenic nuclei serve as diploid controls here, which are essential to confidently gate around the peak that corresponds to the DAPI-stained haploid nuclei (Figure 2f)” [lines 250-252].

4. In the abstract, the Author claim that the method is reproducible, but non data on this point are provided

We have changed the word “reproducible” to “reliable”. We appreciate the word “reproducible” conjures up an expectation of “reproducible phenotypes”. Whereas we mean a “reliable protocol” that works every time, assuming other aspects of quality assurance in the user’s laboratories are met. 

We have included a new Supplementary Table 1 which has data for all experiments that used this technique since it was optimised.

Minor:

Figure 1 is confused with Figure 2

We now refer, accurately, to Figure 2 [line 224].

Referring to real Figure 1 is lacking in the text

We now refer to Figure 1 in a number of appropriate sections [lines 61, 77, 144, 177]. 

Line 195, G1 peak does not seem centred on x-axis.

We modified the text to instruct that the G1 peak needs to be “in range”. This sentence on now reads;

“This G1 peak – of the diploid sample – is placed in-range, visually, on the x-axis by adjusting the voltage.” [lines 224-225]

Reviewer #2: 

The protocol provides a valuable addition to the field of single cell assays of testis tissue, as especially the nucleus isolation has proven tricky in the past. The description of working steps is concise, and I would feel confident following this protocol.

However, I do have some concerns with the manuscript:

1) The authors refer to the method as “single-sperm nucleus isolation”. I find this misleading. Mouse testis contains haploid spermatids at different stages in development and transcription profiles differ distinguishably from round spermatids to later spermatid stages. The nucleus shown in figure 2C corresponds to a later spermatid stage, however FACS sorting of the haploid population would in my opinion also include more immature round spermatids. If this population is excluded in the protocol, it is unclear how. This should be discussed and Figure 2 should be updated to depict not only a single nucleus, but a representative sample of the obtained haploid nuclei.

We have modified the text and a new Figure 3 to illustrate that this protocol is designed to isolate all types of haploid cells from a male, that are available from the given tissue; from round spermatids to mature sperm. We also make clear in the text that the protocol is designed to isolation both spermatids and spermatozoa: 

“This method is developed to isolate nuclei from both spermatids and spermatozoa.” [line 63]

2) In line with my first concern, the authors imply that this protocol can be adapted to isolate mature human sperm. Maturation of sperm, including development of fertilisation capability and motility, is completed not in the testis, but in the epididymides. In my experience, motile sperm is “more sticky” than spermatids and it would be interesting to see how the protocol performs with this population. At least, the distinction between spermatids, spermatozoa and mature sperm should be made clear in the manuscript and discussed in more detail.

We agree that we did not make sufficiently clear that our motivation for developing this protocol was due to our own technical challenges due to the stickiness of these cell types. Therefore, we developed a method that focuses on the isolation of intact nuclei. This is now made explicit and reads:

“A major challenge for the isolation of individual haploid cells from males for sequencing is that they stick to most pipette tips and tubes, and traditional methods for bulk sperm isolation use technically challenging density gradients. Therefore, the motivation for developing this technique is to have a simple and rapid nucleic acid preparation technique that is compatible with current sequencing technologies,…” [lines 42-46].

We also have added new data in Supplementary Table 1, showing how with epididymides we used the protocol to isolate between 320,000 to 1,100,000 haploid nuclei.

3) I was unable to assess the data from the case study, as it was not made available to reviewers prior to publication. Thus, I feel a bit uncomfortable to classify this as a validated method. At least the quality control data for the case study, for example an image of isolated nuclei in the hemocytometer to assess nuclear integrity, should be included.

In light of these points, I recommend the manuscript be revised.

We have included a new Supplementary Table 1 which has data for all experiments that used this technique since it was optimised. We have modified Figure 2 as suggested to show more examples of nuclear integrity. We have also referenced a new manuscript which is on Biorxiv and under review at PNAS, which has used this technique to identify reproducible phenotypic differences between genotypes

(Tsui et al, Figure 1b, https://doi.org/10.1101/2022.06.16.496499).

Minor

(taken from the reviewers’ field: “Is the article presented in an intelligible fashion and written in standard English?”)

Reviewer #1: Yes

Reviewer #2: No: It is a clear and concise description, I noticed only some minor typos:

line 127 – placed

Done [line 151]

lines 151/152 – squeezing the testis with pointed tweezers until the testis bursts

Done [lines 176-177]

line 156 – microcentrifuge tube

Done [lines 183-184]

line 192 – Not a functional sentence

We modified this sentence for clarity, to read; 

Diploid control material – here, the single splenic nuclei suspension – is used to identify the G1 diploid peak based on DAPI intensity, which in turn is used to identify haploid cells from reproductive tissue (Figure 2f). [lines 228-230]

---

## [Decision Letter · Decision Letter 1]

30 Aug 2022

PONE-D-22-14844R1SSNIP-seq: A simple and rapid method for isolation of single-sperm nucleic acid for high-throughput sequencingPLOS ONE

Dear Dr. Crismani,

Thank you for submitting your manuscript to PLOS ONE. After careful consideration, we feel that it has merit but does not fully meet PLOS ONE’s publication criteria as it currently stands. Therefore, we invite you to submit a revised version of the manuscript that addresses the points raised during the review process.

Please include the following items when submitting your revised manuscript:A rebuttal letter that responds to each point raised by the academic editor and reviewer(s). You should upload this letter as a separate file labeled 'Response to Reviewers'.A marked-up copy of your manuscript that highlights changes made to the original version. You should upload this as a separate file labeled 'Revised Manuscript with Track Changes'.An unmarked version of your revised paper without tracked changes. You should upload this as a separate file labeled 'Manuscript'.If applicable, we recommend that you deposit your laboratory protocols in protocols.io to enhance the reproducibility of your results. Protocols.io assigns your protocol its own identifier (DOI) so that it can be cited independently in the future. For instructions see: https://journals.plos.org/plosone/s/submission-guidelines#loc-laboratory-protocols. Additionally, PLOS ONE offers an option for publishing peer-reviewed Lab Protocol articles, which describe protocols hosted on protocols.io. Read more information on sharing protocols at https://plos.org/protocols?utm_medium=editorial-email&utm_source=authorletters&utm_campaign=protocols.

We look forward to receiving your revised manuscript.

Kind regards,

Joël R Drevet, Ph.D.

Academic Editor

PLOS ONE

Journal Requirements:

Additional Editor Comments (if provided):

Although the manuscript has been improved as requested by the reviewer, the addition of new material has raised new questions. Please address the reviewer's concerns.

Reviewers' comments:

Reviewer's Responses to Questions

**Comments to the Author**

1. Does the manuscript report a protocol which is of utility to the research community and adds value to the published literature?

Reviewer #1: Yes

2. Has the protocol been described in sufficient detail?

Descriptions of methods and reagents contained in the step-by-step protocol should be reported in sufficient detail for another researcher to reproduce all experiments and analyses. The protocol should describe the appropriate controls, sample sizes and replication needed to ensure that the data are robust and reproducible.

Reviewer #1: Partly

3. Does the protocol describe a validated method?

Reviewer #1: Yes

4. If the manuscript contains new data, have the authors made this data fully available?

Reviewer #1: Yes

**5. Is the article presented in an intelligible fashion and written in standard English?**

Reviewer #1: Yes

6. Review Comments to the Author

Reviewer #1: The Authors amended the MS according to requests. However, I have some further questions on the newly added material.

Supplemental figure 1. Please better explain: in the “nuclei haploid events” column, there is the final number of aploid nuclei, i.e. that post sorting. In the last column (percentage haploid of population), are the percentages referred to what? To the total haploid nuclei? To the total acquired haploid nuclei? For instance, in the first row, 100,000 nuclei represent the 26,7% of what? How did you count the total haploid nuclei?

In figure 2, the titles of the axes are not readable. Similarly for values in the axes (too small characters). In addition, it is not clear whether the gates were applied one within the other (from A to E) or not (i.e. independently of each other). In particular, the histograms of figures F from which dot plot are generated? This information has to be added.

7. PLOS authors have the option to publish the peer review history of their article (what does this mean?). If published, this will include your full peer review and any attached files.

Reviewer #1: No

---

## [Author Response · Author response to Decision Letter 1]

1 Sep 2022

Dear Dr Drevet

Thank you for sharing reviewer #1’s comments. Reviewer #1 acknowledges that we made the requested improvements. They raise two minor points which are incorporated into this revised version of the manuscript and are described concisely below. 

(please see the attached word document for the same responses as below, but with better formatting)

Reviewer #1: The Authors amended the MS according to requests. However, I have some further questions on the newly added material.

Supplemental figure 1. Please better explain: in the “nuclei haploid events” column, there is the final number of aploid nuclei, i.e. that post sorting. In the last column (percentage haploid of population), are the percentages referred to what? To the total haploid nuclei? To the total acquired haploid nuclei? For instance, in the first row, 100,000 nuclei represent the 26,7% of what? How did you count the total haploid nuclei?

Response - We have expanded the legend of S1 Table to explain what these terms represent. Further we have improved the terms used. We have redefined “nuclei haploid events” to be “gated haploid nuclei”. Similarly, “percentage haploid of population” was changed to “haploid percent of all events pre-gating”

The new S1 Table legend now reads:

“S1 Table. Yields of intact single haploid nuclei with the protocol: 

Yield data from our laboratory using the protocol since optimisation. The data is from fresh and frozen testes, fresh epididymides, and cover wild-type and hypospermatogenic (Fancm-deficient) mice. “gated haploid nuclei” represents the total number of haploid nuclei that were gated using the strategy shown in Fig 2A-F and isolated at gate “R6” (Fig 2F) . “haploid percent of all events pre-gating” represents the number of haploid nuclei were gated (R6, Fig 2F) for a given experiment divided by the total number of events detected pre-gating. i.e. (R6 / events pre-gating) X 100.”

Reviwer #1 - In figure 2, the titles of the axes are not readable. Similarly for values in the axes (too small characters). In addition, it is not clear whether the gates were applied one within the other (from A to E) or not (i.e. independently of each other). In particular, the histograms of figures F from which dot plot are generated? This information has to be added.

Response - We agree and we have made these suggested changes. We have shown that the gating strategy from Fig 2A to 2F was applied sequentially to gated events from the previous step, both on the figure with arrows, and specified in the figure legend. Further, the axis titles have been increased as requested, as have the numbers on the axes. As the intensity numbers (0-250K) are arbitrary we removed the numbers from 50K-200K to create more space for larger font for the minimum and maximum values, but explain in the Fig 2 legend the scale that the major and minor ticks represent 50K and 10K respectively. We also upload an electronic vectorized version of the image that will ensure high resolution can be rendered into the published version. 

The new Figure 2 legend now reads:

“Fig 2. Flow cytometric gating strategy for the isolation of haploid cells. 

The following gating steps are applied sequentially, as indicated by the arrows i.e. all events in ‘B’ represent only the gated events from ‘A’ and so on to ‘F’. (A) Nuclei are isolated from the rest of the population initially using linear forward scatter area (FSC-A-lin) and linear side scatter area (SSC-A-lin) by region gate 1 (R1). (B) Single nuclei are isolated using: region gate 2 (R2) on linear side scatter width (SSC-W-lin) and linear side scatter height (SSC-H-lin); (C) region gate 3 (R3) on linear forward scatter width (FSC-W-lin) and linear forward scatter height (FSC-H-lin); (D) region gate 4 (R4) on linear DAPI area (DAPI-A-lin) and linear DAPI width (DAPI-W-lin); and (E) region gate 5 (R5) on linear DAPI area (DAPI-A-lin) and linear DAPI height (DAPI-H-lin) (F) The use of a diploid control allows flow cytometric gating of haploid sperm. Diploid samples such as splenic cells used here, will have a G1 (2c DNA content) and a G2 peak (4c DNA content, out of range in this figure) peak respectively. Acquire events from region 6 (R6), the haploid nuclei which have a 1c DNA content (R6) which are not present in the diploid control sample. The values on all both axes of panels A-E and the x-axis of panel F are arbitrary values that represent increasing intensity of signal from 0, the lowest, to 250K the greatest. Major and minor ticks represent 50K and 10K increments respectively.”

---

## [Editor Report · Decision Letter 2]

12 Sep 2022

SSNIP-seq: A simple and rapid method for isolation of single-sperm nucleic acid for high-throughput sequencing

PONE-D-22-14844R2

Dear Dr. W.  Crismani,

We’re pleased to inform you that your manuscript has been judged scientifically suitable for publication and will be formally accepted for publication once it meets all outstanding technical requirements.

Kind regards,

Joël R Drevet, Ph.D.

Academic Editor

PLOS ONE

---

## [Editor Report · Acceptance letter]

19 Sep 2022

PONE-D-22-14844R2 

SSNIP-seq: A simple and rapid method for isolation of single-sperm nucleic acid for high-throughput sequencing 

Dear Dr. Crismani:

I'm pleased to inform you that your manuscript has been deemed suitable for publication in PLOS ONE. Congratulations! Your manuscript is now with our production department. 

Kind regards, 

on behalf of

Prof. Joël R Drevet 

Academic Editor

PLOS ONE